# Sublingual Microcirculation Specificity of Sickle Cell Patients: Morphology of the Microvascular Bed, Blood Rheology, and Local Hemodynamics

**DOI:** 10.3390/ijms24043621

**Published:** 2023-02-11

**Authors:** Sachi Sant, Etienne Gouraud, Camille Boisson, Elie Nader, Mounika Goparaju, Giovanna Cannas, Alexandra Gauthier, Philippe Joly, Céline Renoux, Salima Merazga, Christophe Hautier, Philippe Connes, Marianne Fenech

**Affiliations:** 1Department of Mechanical Engineering, University of Ottawa, Ottawa, ON K1N 6N5, Canada; 2Laboratoire Interuniversitaire de Biologie de la Motricité (LIBM) EA7424, Team «Vascular Biology and Red Blood Cell», Université Claude Bernard Lyon 1, 69008 Lyon, France; 3Laboratoire d’Excellence du Globule Rouge (Labex GR-Ex), PRES Sorbonne, 75015 Paris, France; 4Service de Biochimie et Biologie Moléculaire, Laboratoire de Biologie Médicale Multi-Site, Hospices Civils de Lyon, 69008 Lyon, France; 5Service de Médecine Interne, Hôpital Edouard Herriot, Hospices Civils de Lyon, 69008 Lyon, France; 6Institut Hématologique Oncologique Pédiatrique (IHOPe), Hospices Civils de Lyon, 69008 Lyon, France

**Keywords:** sickle cell disease, microcirculation, red blood cell deformability, sidestream dark field imaging

## Abstract

Patients with sickle cell disease (SCD) have poorly deformable red blood cells (RBC) that may impede blood flow into microcirculation. Very few studies have been able to directly visualize microcirculation in humans with SCD. Sublingual video microscopy was performed in eight healthy (HbAA genotype) and four sickle cell individuals (HbSS genotype). Their hematocrit, blood viscosity, red blood cell deformability, and aggregation were individually determined through blood sample collections. Their microcirculation morphology (vessel density and diameter) and microcirculation hemodynamics (local velocity, local viscosity, and local red blood cell deformability) were investigated. The De Backer score was higher (15.9 mm^−1^) in HbSS individuals compared to HbAA individuals (11.1 mm^−1^). RBC deformability, derived from their local hemodynamic condition, was lower in HbSS individuals compared to HbAA individuals for vessels < 20 μm. Despite the presence of more rigid RBCs in HbSS individuals, their lower hematocrit caused their viscosity to be lower in microcirculation compared to that of HbAA individuals. The shear stress for all the vessel diameters was not different between HbSS and HbAA individuals. The local velocity and shear rates tended to be higher in HbSS individuals than in HbAA individuals, notably so in the smallest vessels, which could limit RBC entrapment into microcirculation. Our study offered a novel approach to studying the pathophysiological mechanisms of SCD with new biological/physiological markers that could be useful for characterizing the disease activity.

## 1. Introduction

Sickle cell disease (SCD) is an inherited red blood cell disorder caused by a single mutation in the β-globin gene. Due to this mutation, abnormal hemoglobin (called HbSS) may polymerize when deoxygenated, leading the biconcave disc red blood cell (RBC) shape to change into that of a crescent moon. RBCs in individuals suffering from SCD are known to be more rigid [1] and sticky [2], hence promoting vaso-occlusion. As a result of these severe hemorheological alterations, functional blood flow to vital organs and other bodily structures is compromised. Individuals suffering from vascular obstruction may experience side effects varying from mild pain to organ failure. Although gene therapy and hematopoietic stem cell transplants are currently under trial for SCD, [3,4,5] the main treatments available are those that limit the complications and pain associated with the disease [1]. Because the RBCs in SCD patients are more fragile than those in healthy individuals, patients also develop chronic hemolytic anemia with a hemoglobin level of around 7–9 g/dL. The improvement and advancement of therapies aimed at tackling SCD start with understanding the behavior of the disease within an individual.

SCD is also characterized by chronic vascular dysfunction due to the effects of free heme and hemoglobin on endothelial cells and inflammation [6]. It has been demonstrated that free hemoglobin may scavenge nitric oxide, hence decreasing the bioavailability of this compound and decreasing the vasomotor reserve [6]. In addition, the release of arginase from RBCs during hemolysis results in the consumption of L-arginine (i.e., the precursor to nitric oxide), further decreasing nitric oxide bioavailability [6]. Moreover, free heme may promote inflammation through the activation of several pathways, such as the NF-κB and NLRP3 inflammasome pathways [6]; indeed, free heme is now considered to be an erythroid damage-associated molecular pattern (eDAMP) [6]. The activation of endothelial cells also results in the overexpression of adhesion molecules from the selectin and super immunoglobulin families, leading to an increase in the adhesion of all circulating cells (white blood cells, platelets, and RBCs) to the endothelial wall [6]. Increased cell adhesion to the vascular wall may slow down the velocity of RBCs, which would eventually spend more time in deoxygenated areas, thus increasing the risk of sickling and complete vascular occlusion occurring if the vascular system is not able to adapt [6].

The overall function of microcirculation is critical for supporting organ activity and proper functioning [7]. Microcirculation is the main site of oxygen exchange from the RBCs to the tissues in the vascular system and is additionally responsible for the exchange of solutes, hormones, and nutrients; thus, it is critical for understanding the flow behavior within these sites [7]. Tissue perfusion under changing conditions can be investigated using intravital microscopy. Typically, studies seek to document the spatial and temporal variability of microcirculation, as it affects perfusion [8]. Microcirculation in the nail, cornea, or sublingual area can be accessed noninvasively, while microcirculation in other locations, such as the brain, can be studied using endoscopies or during surgery. The main parameters extracted from these analyses are geometrical parameters (vessel diameters and density) and local velocities. Sublingual microcirculation has been studied in a few diseases, such as pulmonary arterial hypotension and systemic sclerosis [9,10]. Van Beers et al. previously investigated the blood velocity in sublingual microcirculation during a painful crisis in SCD patients [11]. The role of hemorheological abnormalities in the causation of several SCD complications is the focus of recent studies [12]. It has been demonstrated that the blood viscosity and RBC deformability are lower in SCD patients compared to controls [13,14,15]. The reduction in RBC deformability in SCD patients is due to the presence of the less soluble HbSS polymers, rather than normal hemoglobin, in the cytosol; the high internal viscosity due to cell dehydration; and the loss of membrane elasticity due to the accumulation of membrane damages [12]. Despite the reduction in RBC deformability, the blood viscosity of SCD patients is lower than that of healthy individuals because the hematocrit is very low in SCD patients (20–25% versus 40–50% in SCD and healthy individuals, respectively). Nevertheless, any rise in the blood viscosity in SCD patients may precipitate the onset of a vaso-occlusive crisis [12]. However, the blood rheological properties are always measured ex vivo in blood samples, and no current technique is able to directly measure both the microcirculation dynamics and blood rheology in humans in vivo. Blood is a shear-thinning fluid, and RBC deformability may vary with the geometrical and flow conditions of the vessels, which may be different from one individual to another [16,17]. Studies exploring the density and morphology of microcirculation through muscle biopsies in SCD patients showed a lower capillary density, a greater tortuosity, and larger capillaries in this population compared to healthy controls [18], but the impact of such differences on the blood flow dynamics and on the local blood rheology is unknown.

The present study aimed to investigate the systemic vascular changes (vessel density) associated with the local microhemodynamic (velocity) and local hemorheological properties (viscosity and RBC deformability) in individuals with SCD. The vessel diameters, local velocities, and blood rheological parameters were integrated to further characterize the microhemodynamics and local hemorheological properties in individuals with SCD. To support the analysis of the sublingual microcirculation, a blood rheological analysis was performed individually on blood samples [19]. Ex vivo subject-specific rheological properties were then used to derive the local in vivo flow properties to reflect the physiological flow conditions within the vessels.

## 2. Results

### 2.1. Blood Rheology Measured Ex Vivo

There were significant differences between the patients with sickle cell anemia (HbSS) and the healthy individuals (HbAA) with respect to the hematocrit (*p* < 0.001), the viscosity flow consistence index k (*p* < 0.05), and RBC deformability (*p* < 0.001), which were found to be higher in the HbAA individuals (Table 1).

### 2.2. Sublingual Microcirculation Profile

The De Backer Score was higher (*p* < 0.01) in the HbSS subjects compared to the HbAA subjects (Figure 1a). The analysis of variance indicated that the distribution of the vessel diameters was different between the HbAA and HbSS groups (Figure 1b; *p* < 0.05). The HbSS volunteers had a higher number of vessels with diameters < 6 µm than the control volunteers (*p* < 0.05). In contrast, the HbAA group exhibited a significantly higher proportion of vessels > 16 µm (*p* < 0.05). The greatest proportion of microvessels in the HbSS group was in the 6–8 µm range, while it was in the 8–10 µm range for the HbAA group. Then, we analyzed the blood flow velocity (Figure 1c) and shear rate (Figure 1d) values in the two groups. Because the velocities were sporadically detectable in the smallest vessels (2–4 µm), this vessel group was removed from the analysis due to a lack of available data. Notably, in 93 ± 11% of the analyzed sites, no 2–4 µm vessels were flowing in the HbSS group compared to 70 ± 23% in the HbAA group (*p* = 0.22). The velocities of the HbSS and HbAA individuals were not significantly different (*p* = 0.104). The shear rate of the two groups decreased as the vessel diameter increased (*p* < 0.001) and tended to be slightly higher in the HbSS group than in the HbAA group (*p* = 0.054), notably so in the smallest vessel sizes ranging from 6 to 14 µm.

### 2.3. Local Blood Rheology

The viscosity in the HbSS volunteers was determined to be significantly lower than that in their HbAA counterparts (*p* < 0.001) across almost all the vessel diameters with a detailed distribution outlined in Figure 2a. Because the shear rate decreased when the vessel diameters increased, the blood viscosity increased from the smallest to the biggest vessels in the two groups (*p* < 0.001). No significant difference was found when comparing the shear stress across the vessel diameters of the HbAA group with that of the HbSS group (Figure 2b; *p* = 0.118). The magnitude of the changes in the shear stress between the smallest and the biggest vessels was comparable between the HbAA group (74.3% decrease in the 4–6 µm and 26–28 µm categories) and the HbSS group (71.3% decrease in the 4–6 µm and 26–28 µm categories). Figure 2c shows the changes in local RBC deformability across all the vessel diameters in the two groups. RBC deformability was higher in the healthy volunteers compared to the HbSS individuals for the vessel diameters ranging from 4 to 22 μm. Moreover, the increase in RBC deformability between the biggest and the smallest vessels was higher in the HbAA group than in the HbSS group, where RBC deformability remained very low (less than 0.1 a.u.).

## 3. Discussion

Our study demonstrated a greater De Backer score in the HbSS group than in the HbAA group, suggesting a greater capillary density in the former group. The De Backer score encompasses the arterioles, capillaries, and venules. While it does not provide the same information as the functional capillary density, which specifically reflects the metabolic exchange potential and which includes arterioles, it does provide an indication of the density and perfusion of the microvascular bed [20]. This result contrasted with the previous findings obtained through muscle biopsies in SCD patients and showed a rarefaction in the capillaries [18]. The prevalence of no-flow vessels in the HbSS population could reflect pathophysiologic information, as this may impact perfusion/tissue oxygenation; however, there was no statistical difference between the two groups, probably because of the limited sample size. Studies including more patients are clearly needed. In addition, we observed a higher number of small vessels and a lower proportion of large vessels in the HbSS group compared to the HbAA group, which also contrasted with the muscle biopsies findings, showing a higher number of large capillaries in the SCD patients than in the healthy controls [18]. One of the differences between our study and the study of Ravelojaona et al. [18] was that the analyses of microcirculation were not done in the same tissue: sublingual vs. muscle. Moreover, Merlet et al. [21] recently reported that eight weeks of an exercise training program increased the capillary density of SCD patients. Indeed, high interindividual variability may exist in the SCD population depending on the severity of the disease and on a more or less sedentary lifestyle. Nevertheless, other diseases causing inefficient oxygen transport, such as Moyamoya disease, have been found to have increased vascularization with an increased vessel density compared to control populations [22].

The denser microcirculation network and the increase in the smaller vessels observed in the HbSS population may be credited to the fact that SCD individuals suffer from repeated ischemic episodes due to anemia and repeated microcirculatory blockages due to their poorly deformable RBCs [23]. New blood vessels usually develop in places where they are most needed [24], and the serum levels of the angiogenic factors have been found to be elevated in SCD patients, which indicates a proangiogenic state in this disease [25]. In response to hypoxia, the body tries to counteract this through an adaptive response by developing more transport methods for the RBCs to reach optimum oxygen levels, hence causing an increase in vessel density, a phenomenon called hypoxia-induced angiogenesis [25]. This may occur to maintain organ integrity and an adequate vascular system in cases where blood transport is compromised. Park et al. observed pathologic angiogenesis in the bone marrow of SCD mice associated with highly tortuous arterioles and increased HIF-1α levels [26]. This phenomenon was also observed in other studies that examined an increased vessel density and its impact on blood oxygen transfer in hypoxic environments [27,28,29,30].

A previous study on sublingual microcirculation in SCD individuals showed that the sublingual microcirculatory blood flow velocity was not impaired in SCD patients during a painful crisis [11]. Our study also showed that the sublingual blood flow velocity in the HbSS patients that were in a steady state (i.e., not in crisis) was not different from that of the healthy controls despite the fact that the hematocrit and blood viscosity were lower in the HbSS individuals. Nevertheless, the sample size of our HbSS group was very small, and it seems that the blood flow velocity tended to be slightly higher in the HbSS group than in the healthy individuals, notably so in the smallest vessels, which could limit RBCs to being trapped in microcirculation. Veluswamy et al. reported that fast-moving RBCs are better able to escape microcirculation before the polymerization of abnormal hemoglobin occurs [31], hence preventing vaso-occlusive crises.

The shear stress of all the vessel diameters was not different between the HbSS patients and the healthy individuals because the shear rate tended to be higher and because the blood viscosity was lower in the HbSS patients compared to the healthy controls. However, RBC deformability was very low in the HbSS individuals compared to the control group, notably so in the smallest vessels. Indeed, in the case of a further loss of vascular reactivity, the accumulation of poorly deformable RBCs would precipitate the onset of vaso-occlusive-like complications. In vessels larger than 22 μm, RBC deformability was not different between the two populations because the shear stress was very low and very similar between the two groups. The low RBC deformability in these vessels was less problematic because the vessel diameter was almost three-fold the diameter of the RBCs, which would allow them to flow more easily than they would in the smallest vessels.

Given the variability of the HbAA and HbSS data in the study and the small population size, the results may not be definitive, and several points deserve closer attention in future studies. The clinical relevancy of the qualitative flow in the microvessels (continuous, sluggish, stop and go, or no flow) could be further tested, as it can reflect pathophysiological information. The correlation between the estimated local shear stress and local autoregulation with endogenous nitric oxide and other endothelial autocoids would also be of great interest. Such correlations could open the door to personalized therapeutic management using a point-of-care sublingual microcirculation assessment. Several other diseases where the RBC rheology and microcirculation are impaired, such as thalassemia, hereditary spherocytosis, channelopathies, metabolic diseases, etc., could benefit from such an integrative approach, which would improve clinical management and identify novel therapies.

## 4. Materials and Methods

### 4.1. Subjects

As part of a wider study presented in [32], sublingual video microscopy recordings were obtained for 8 HbAA subjects (27–42 yrs, 4 males/4 females) and 4 HbSS patients (15–50 yrs, 2 males/2 females) from the University Hospital of Lyon (Hospices Civils de Lyon, Lyon, France). Patients accepted into the trial were first analyzed to ensure that participation requirements were met, which included them being in a clinical steady state. This included no acute vaso-occlusive crises, acute chest syndrome, history of hospitalization in the previous 2 months, or history of blood transfusion within 3 months prior to the study. Furthermore, this study was performed as per the Declaration of Helsinki guidelines and was approved by the French Ethics Committee (CPP Est IV, Strasbourg, France; clinical trial number: NCT03243812). Written informed consent was collected from all volunteers in the study, with parent consent obtained for those under 18 years old. Blood was sampled in EDTA tubes for the measurement of blood rheological parameters.

RBC deformability and blood viscosity measurements were performed in the hour following the sampling. The EDTA blood samples were oxygenated at room air temperature for 15 minutes right before ex vivo analysis.

### 4.2. RBC Deformability

RBC deformability was determined at 37°C and at shear stress ranging from 0.3 to 30 Pa through ektacytometry (Laser-Assisted Optical Rotational Cell Analyzer, LORRCA MaxSis, RR Mechatronics, Hoorn, Netherlands) [33]. RBCs were suspended in polyvinylpyrrolidone (PVP, viscosity = 30 cP) and were placed between two concentric cylinders (Couette system) as shear stress was applied to generate a thin sheared layer. The RBCs were deformed and then the shape of the diffraction pattern obtained using laser beam diffraction was changed. The diffraction patterns were circular at low shear and became elliptical with increased shear stress. The elliptical shape could be obtained because the external viscosity (of the PVP) was greater than the internal viscosity of the cells. The elongation index (*EI*) could, thus, be calculated based on the geometry of the ellipse, using the formula *EI* = (length − width)/(length + width). A higher *EI* was reflective of a higher ability of the RBCs to deform. The elongation index–shear stress curves were subsequently parameterized with a Lineweaver–Burke relation. The fitted function thus obtained gave the personalized relation presented by Equation (1):(1)$$EI_{LB}\left(SS\right)=\frac{EI_{max}\;\times \;SS}{A+SS},$$
where *EI_max_* is the maximum theoretical deformation, *SS* is shear stress, and *A* is a constant parameter of the slope of 1/*EI* vs. 1/*SS* plot. A sample curve fit for a volunteer with sickle cell anemia (HbSS) is presented in Figure 3a.

In vivo, local RBC deformability in the microcirculation network was then estimated using Equation (1) to obtain the corresponding local shear stress.

### 4.3. Blood Viscosity

Blood viscosity was measured for native Hct using a cone/plate viscometer (Brookfield DVII + with CPE40 spindle; Brookfield Engineering Labs, Natick, MA, USA) at 2.25, 11.5, 22.5, 45, 90, and 225 s^−1^ and at room air conditions. Using these data, a shear rate–viscosity plot was developed to derive the rheological law for each volunteer. A personalized power law was derived by plotting the viscosity at various shear rates measured using the viscometer. Power law of non-Newtonian fluids, such as blood [34], takes form as outlined in Equation (2) [35].
(2)$$\mu =k\;\times \;\gamma^{\left(n-1\right)},$$
where *μ* is viscosity, *k* is the flow consistence index, *γ* is the shear rate, and *n* is the power law index. A sample curve fit for a volunteer with sickle cell anemia (HbSS) is presented in Figure 3b. The derived power law was then used to conduct local parameter estimates for the local in vivo shear rates.

### 4.4. Live Imaging of Microcirculation: Intravital Microscopy and Postprocessing

Sublingual microcirculation was analyzed using sidestream dark field (SDF) imaging (Microscan, MicroVision, Amsterdam, The Netherlands). In brief, imaging of sublingual microcirculation was performed at the site of interest. SDF imaging utilizes a concentric arrangement of light-emitting diodes (LEDs) to enhance contrast and to reduce blurry imaging [36]. The resulting image is a gray and white video with dark moving speckles representing RBC flow. The light from the device is absorbed within the RBCs, causing the dark and high-contrast spheres [37]. The De Backer score, the diameter distribution, and the velocities were determined using the Automated Vascular Analysis software (AVA, Microscan, MicroVision, Amsterdam, The Netherlands). For each volunteer, videomicroscopy recordings of three to six sublingual sites were analyzed. The sites analyzed were determined based on the video quality obtained, as patients found it difficult to hold their tongue still. Furthermore, to avoid data inaccuracies, the operator maintained constant pressure on the Microscan handheld device, which requires dexterity and practice. The video of each site was approximately 10 s long, and the most stable sequence of frames was extracted for analyses.

The De Backer score is a measurement of the vessel density and is determined by counting the number of vessels crossing arbitrary horizontal and vertical lines [20] as shown in Figure 4. It is calculated by dividing the number of vessels crossing the line by the length of the line.

The microvessels were classified by size and were placed in 14 groups from 2–4 µm to 28–30 µm in diameter. The proportion of vessels in each group size was characterized by the proportion of cumulative length. The proportion of cumulative length is defined by the AVA software used as the sum of the lengths of the vessels in the given group size divided by the total length of all the vessels of all sizes included. For each individual, the data from the different recordings were cumulated.

The local velocity in each vessel was determined using AVA software based on kymographs. Kymographs are plots representing spatial position as a function of the time used to quantify velocity along a determined path [38]. Kymographs were plotted for each detected vessel along the centerline [8]. The resulting velocity vectors were individually visually validated by the operator. For each subject, the velocities were averaged for each vessel group size.

### 4.5. Estimation of Local Blood Viscosity, Shear Rate, Shear Stress, and RBC Deformability

In each group size, the diameters and velocities obtained as described previously were used to further estimate the local shear rate γ using the Hagen–Poiseuille relation in circular channels [35] as presented in Equation (3).
(3)$$\gamma =\frac{4\;\times \;V}{3\;\times \;D}\;,$$
where *V* is the velocity and where *D* is the vessel diameter.

Then, the personalized law of viscosity from Equation (2) was used in combination with the local shear rate (Equation (3)) to determine local shear stress (*SS*) [35]:(4)SS=μ × γ,
where μ is viscosity and γ the shear rate.

Finally, RBC deformability (i.e., *EI*) exhibited by each subject at the different vessel sizes was estimated using the personalized elongation index relationship in Equation (1) to determine the shear stress of the given vessel size.

Figure 5 outlines the process undertaken to derive the local parameter estimations for blood rheology characterization of healthy and SCD individuals.

### 4.6. Statistical Analyses

An unpaired Student’s t test was used for comparisons between the two groups. A one-way analysis of variance (ANOVA) was used to test the changes in several parameters across all vessel diameters. To perform the statistical tests, Minitab data analysis software was used, and GraphPrism 9 was used for the graphical representations. A *p* value less than 0.05 was considered to be statistically significant.

## 5. Conclusions

This study was the first to integrate local in vivo microcirculatory hemodynamics and ex vivo blood rheological data to generate local microcirculatory blood rheological information in humans, and, more particularly, in SCD, a disease characterized by vascular dysfunction and severe rheological blood alterations. Although the sample sizes of the groups were limited, our study offered a novel approach to studying the pathophysiological mechanisms of SCD, and it would have to be tested in other situations (steady state vs. vaso-occlusive crisis; effects of sickle cell genotypes; effects of treatments, such as hydroxyurea or voxelotor treatments; effects of simple transfusion or exchanged transfusion; associations with some specific complications involving strong vascular components, such as cerebral vasculopathy, leg ulcers, etc.) to see if it can offer relevant biological and physiological markers of disease progression to help physicians in managing this disease.

## Figures and Tables

**Figure 1 ijms-24-03621-f001:**
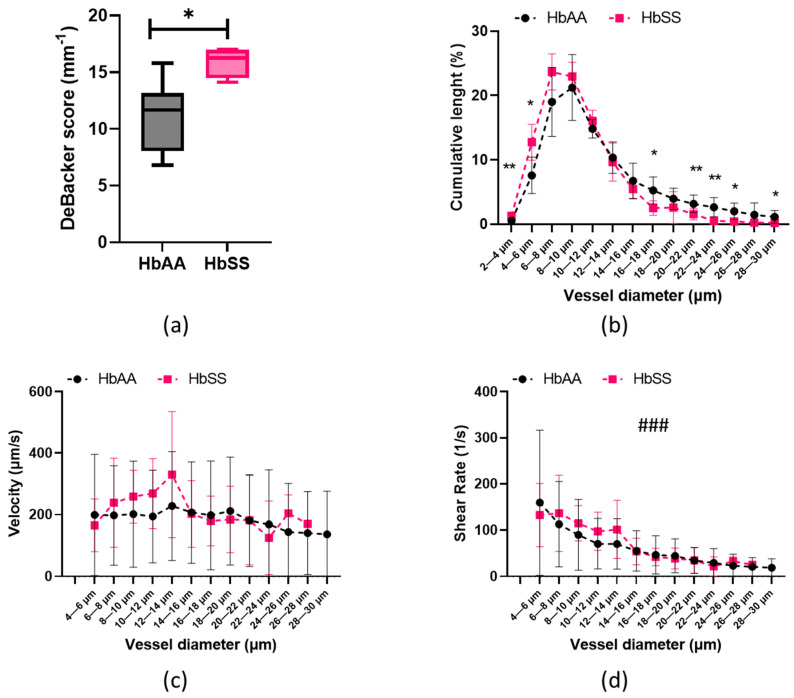
(**a**) De Backer score, (**b**) vessel diameter distribution in sublingual microcirculation, (**c**) velocity distribution within microcirculation, (**d**) and shear rate distribution within microcirculation for HbAA and HbSS groups. Significant differences between HbSS with HbAA groups: * represents *p* < 0.05, and ** represents *p* < 0.01. Significant decrease across the vessel diameters: ### represents *p* < 0.001. No significant difference was found between HbSS group and HbAA group, but a trend was observed of HbSS group having slightly higher shear rates (*p* = 0.054).

**Figure 2 ijms-24-03621-f002:**
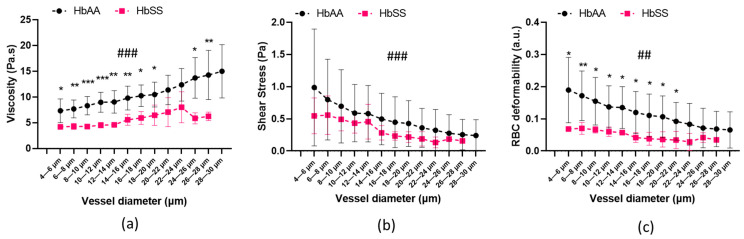
(**a**) Local viscosity in sublingual microcirculation, (**b**) local shear stress in sublingual microcirculation, (**c**) RBC deformability (EI) distribution in sublingual microcirculation in HbAA and HbSS individuals. Significant differences between HbSS and HbAA groups: * represents *p* < 0.05, ** represents *p* < 0.01, and *** represents *p* < 0.001. Significant changes in microcirculation (over vessel diameters): ## represents *p* < 0.01, and ### represents *p* < 0.001.

**Figure 3 ijms-24-03621-f003:**
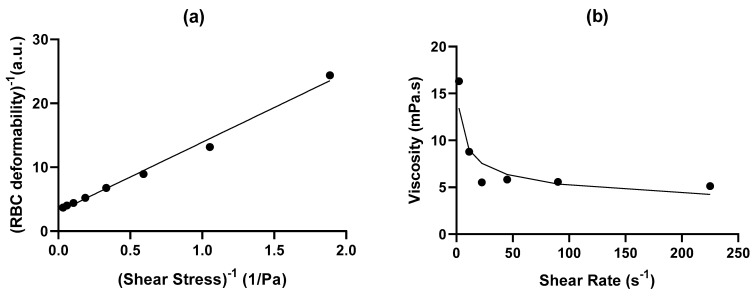
Example of curve fitting for a volunteer with sickle cell anemia (HbSS) with (**a**) Lineweaver–Burke relationship, i.e., inverse deformation as a function of inverse shear stress leading to the personalized function presented in Equation (1), and (**b**) viscosity as a function of the shear rate fitted with power law (Equation (2)).

**Figure 4 ijms-24-03621-f004:**
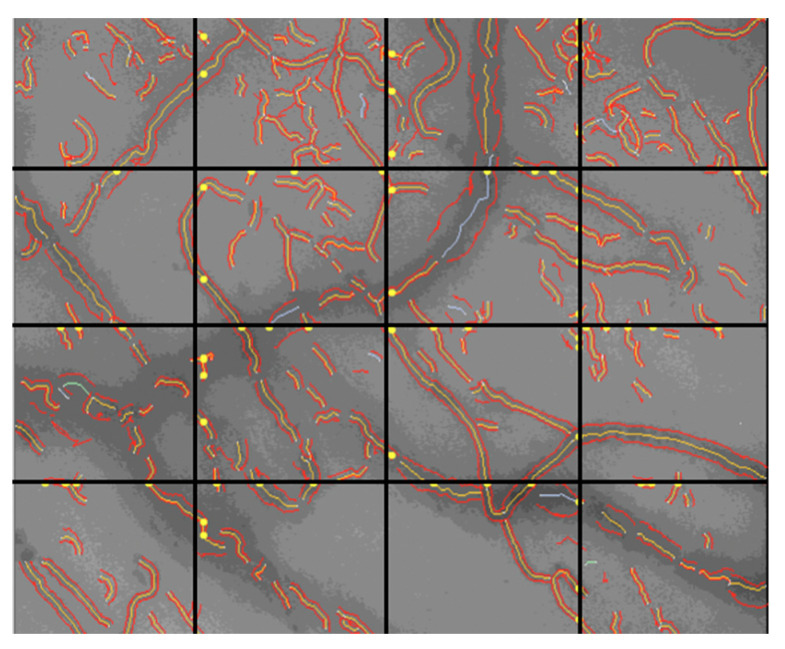
Determination of De Backer Score with vessel segmentation extracted with the AVA software.

**Figure 5 ijms-24-03621-f005:**
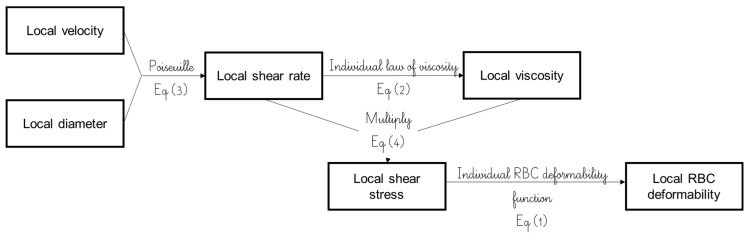
Blood rheological characterization flowchart using individual parameter estimations of each vessel size.

**Table 1 ijms-24-03621-t001:** Blood rheological parameters for HbAA and HbSS. Significant difference between the two groups: * represents *p* < 0.05, and ** represents *p* < 0.01.

Parameter	HbAA	HbSS
Hematocrit (%)	42.1 ± 4.6	29.5 ± 4.0 **
Maximum RBC deformability, EImax (a.u.)	0.65 ± 0.06	0.43 ± 0.09 **
Viscosity flow consistence index, k (Pa.s^n^)	34.4 ± 15.2	16.2 ± 4.2 *
Viscosity power law index, n (a.u.)	0.69 ± 0.07	0.43 ± 0.09

## Data Availability

Data will be made available upon a reasonable request.

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
