# Peer review of "Sublingual Microcirculation Specificity of Sickle Cell Patients: Morphology of the Microvascular Bed, Blood Rheology, and Local Hemodynamics"

_ijms, 2023, doi:10.3390/ijms24043621_

Round 1

Reviewer 1 Report

This study compares microcirculatory sub-lingual micro-hemodynamics of normal (HbAA) vs. sickle cell (HbSS) adult patients showing that the principal difference between groups centers on their respective DeBecker scores, which is significantly higher in Hbss patients. 

It might be advisable to indicate that the DeBecker score includes arterioles, capillaries, and venules and does not provide the same information as the par ammeter termed “functional capillary density” is specific to the capillaries, and their unique ability to extract by-products of metabolism from the tissue.

Notably, local microhemodynamic parameters combine in such a fashion that shear stress tends to be the same in both populations suggesting that in the absence of the sickle cell crisis, local autoregulation by endogenous nitric oxide (NO) and other endothelial autocoids is normal.   This result might not be definitive given the large variability in HbAA data compared with the HbSS data in the study.

The use of asterisks (*) instead of the multiplication symbol (x) in equations 1,2,3 and 4 is not advisable… and the multiplication symbol (x) should be used.

Author Response

Reviewer 1

- This study compares microcirculatory sub-lingual micro-hemodynamics of normal (HbAA) vs. sickle cell (HbSS) adult patients showing that the principal difference between groups centers on their respective DeBecker scores, which is significantly higher in Hbss patients.

It might be advisable to indicate that the DeBecker score includes arterioles, capillaries, and venules and does not provide the same information as the par ammeter termed “functional capillary density” is specific to the capillaries, and their unique ability to extract by-products of metabolism from the tissue.

We thank the reviewer for his/her comment. Changes have been done into the manuscript. “DeBacker score encompasses arterioles capillaries, and venules. While it does not provide the same information as the "functional capillary density”, which specifically reflects the metabolic exchange potential includes arterioles, it does provide an indication of the density and perfusion of the microvascular bed (De Backer et al, 2007).”

- Notably, local microhemodynamic parameters combine in such a fashion that shear stress tends to be the same in both populations suggesting that in the absence of the sickle cell crisis, local autoregulation by endogenous nitric oxide (NO) and other endothelial autocoids is normal.   This result might not be definitive given the large variability in HbAA data compared with the HbSS data in the study.

We agree with the reviewer and added a sentence in the revised manuscript in the perspectives, at the end of discussion: “The correlation between estimate local shear stress and the local autoregulation by endogenous nitric oxide (NO) and other endothelial autocoids would also be of great interest.” 

- The use of asterisks (*) instead of the multiplication symbol (x) in equations 1,2,3 and 4 is not advisable… and the multiplication symbol (x) should be used.

Corrections have been done.

Reviewer 2 Report

In the manuscript entitled "Sublingual microcirculation specificity of sickle cell patients: morphology of the microvascular bed, blood rheology and local hemodynamics" Sachi Sant et al. investigate systemic vascular changes and hemorheological properties in the context of SCD. The authors provide interesting results derived from a novel “paired” in vivo/ex vivo approach to extract information regarding blood rheology. The study is focused, and the authors present an easy to read manuscript.

This Reviewer has only some comments, as follow:

1.       Figures 1 and 2: Some marks of significance (*) are not so easy to read. Could the authors please enlarge them?

2.       Figure 1d: Please replace “sheare” with “shear” in the y-axis. A slight proof-reading for such minor mistakes throughout the manuscript might be needed.

3.       Discussion section: The authors state that “Indeed, high inter-individual variability may exist in the SCD population, depending on the severity of the disease and on the more or less sedentary lifestyle.” This reviewer agrees that this is a major issue when dealing with a specific patient group. First of all, do the authors believe that their proposed study plan could give the potential to “sub-categorize” SCD patients and thus provide information for a more personalized therapeutic protocol/management (in the near future)? This reviewer believes that the manuscript would benefit if such “discussion” is added by the authors along with some more details regarding inter-individual differences in the SCD context (blood vessels, rheology etc.). Finally, can the authors comment if this approach could be extended to other patient groups (thalassemic etc.) as well? This would strengthen and widen the impact of their study results.

Author Response

Reviewer 2

- In the manuscript entitled "Sublingual microcirculation specificity of sickle cell patients: morphology of the microvascular bed, blood rheology and local hemodynamics" Sachi Sant et al. investigate systemic vascular changes and hemorheological properties in the context of SCD. The authors provide interesting results derived from a novel “paired” in vivo/ex vivo approach to extract information regarding blood rheology. The study is focused, and the authors present an easy to read manuscript.

We sincerely thank the reviewer for his/her positive comments.

- This Reviewer has only some comments, as follow:

  1. Figures 1 and 2: Some marks of significance (*) are not so easy to read. Could the authors please enlarge them?

Changes have been done.

  1. Figure 1d: Please replace “sheare” with “shear” in the y-axis. A slight proof-reading for such minor mistakes throughout the manuscript might be needed.

We did the correction.

  1. Discussion section: The authors state that “Indeed, high inter-individual variability may exist in the SCD population, depending on the severity of the disease and on the more or less sedentary lifestyle.” This reviewer agrees that this is a major issue when dealing with a specific patient group. First of all, do the authors believe that their proposed study plan could give the potential to “sub-categorize” SCD patients and thus provide information for a more personalized therapeutic protocol/management (in the near future)? This reviewer believes that the manuscript would benefit if such “discussion” is added by the authors along with some more details regarding inter-individual differences in the SCD context (blood vessels, rheology etc.). Finally, can the authors comment if this approach could be extended to other patient groups (thalassemic etc.) as well? This would strengthen and widen the impact of their study results.

We thank the reviewer for his/her very interesting comment. We added such a paragraph in the discussion: “Given the variability in HbAA and HbSS data in the study and the small population size, results may not be definitive, and several points deserve closer attention in future studies. The clinical relevancy of qualitative flow in microvessels (continuous, sluggish, stop and go, or no flow) could be further tested as it can reflect pathophysiological information. The correlation between estimate local shear stress and the local autoregulation by endogenous nitric oxide (NO) and other endothelial autocoids would also be of great interest. Such correlations could open the door to personalized therapeutic management using the point of care sublingual microcirculation assessment. Several other diseases where RBC rheology and microcirculation are impaired, such as thalassemia, hereditary spherocytosis, channelopathies… but also metabolic diseases, could benefit for such an integrative approach to improve clinical management and identify novel therapies.“

Reviewer 3 Report

This is an innovative synthesis of local in vivo microcirculatory hemodynamics with the Microscan device and ex vivo blood rheology (LORRCA, cone/plate viscometry).  This approach opens new opportunities for microcirculatory studies to build upon the imaging capability of Microscan imaging.  The estimate that individuals with sickle cell disease have low viscosity due to the anemic hematocrit is mildly surprising and needs confirmatory studies, but this paper makes an excellent contribution to science with its demonstration of the integrative methodology.  

The Discussion paragraphs about DeBacker score and capillary density (lines 136- 163) could mention that the highly tortuous vessels could raise the DeBacker score. It seems that a gridline on Fig 4, two parallel vessels could touch the gridlines 4 times and one vessel with tortuous hairpin turns could also touch the gridline 4 times, with the same DeBacker score but different physiologic significance for oxygen transport and vulnerability to vasoocclusion. 

The Conclusion has only the briefest mention of future studies (lines 289-290).  Please consider adding to the Discussion or the Conclusion more discussion of the limitations of this novel approach and plans for future studies. For example,

1. the ex vivo measurements provide the statistical average properties of a population of RBC, but maybe the pathophysiologic events need emphasis on the subpopulation of the most rigid cells (analogous to the large body of data that subpopulations of sickle RBC have high density vs lower density). The most abnormal sickle RBC would probably preferentially cause vaso-occlusion and so the analysis might need to include measures of the heterogeneity of the rheologic properties.

2. the statistical distributions of shear rate and RBC velocity are probably skewed and maybe some pathophysiologic information could be gained from examining the vessels with the lowest shear rate. The proportion of vessels with "sporadically detectable" velocities (lines 100-101) might be an important pathophysiologic measurement. 

3. Blood sample handling and storage conditions could influence RBC survival, so that the most abnormal RBC could be lost before ex vivo analysis. 

4. temporal variation in perfusion. Dr. Tom Coates group has a body of work on vasomotor regulation and analysis of the frequency of cardiovascular variations in larger vessels and heart rate. Is temporal variation also captured in the microcirculation observations by Microscan?

5. The tongue is not a site for vaso-occlusive complications in clinical experience with sickle cell disease, so the findings . Could Microscan provide microcirculatory imaging in any sites that do have vasoocclusion? 

line 44 "Although gene therapy is" should be "Although gene therapy and hematopoietic stem cell transplant are"

line 94 "repartition of the vessel diameters" might be more clearly termed "distribution of the vessel diameters". 

lines 197-198 - please specify the storage time, oxygen conditions, and temperature before ex vivo analysis, and whether the blood sample remained in EDTA during that time.  These parameters can influence sickle RBC 

line 218-219  - please specify the oxygen conditions for the cone/plate viscometer. Presumably it is room air. 

line 241-242 "video microscopy recordings of three to six sublingual sites were analyzed." What determined the number of sites - quality of image? number of datapoints? the ability of the volunteer to hold the tongue position?  what was the duration of recording - could it capture temporal variation of vasoregulation?

Author Response

Reviewer 3

- This is an innovative synthesis of local in vivo microcirculatory hemodynamics with the Microscan device and ex vivo blood rheology (LORRCA, cone/plate viscometry).  This approach opens new opportunities for microcirculatory studies to build upon the imaging capability of Microscan imaging.  The estimate that individuals with sickle cell disease have low viscosity due to the anemic hematocrit is mildly surprising and needs confirmatory studies, but this paper makes an excellent contribution to science with its demonstration of the integrative methodology. 

We thank the reviewer for his/her positive comments. The lower blood viscosity found in HbSS compared to HbAA has been reported several times in the past. References have been added:

Tripette, J., Alexy, T., Hardy-Dessources, M.D., Mougenel, D., Beltan, E., Chalabi, T., Chout, R., Etienne-Julan, M., Hue, O., Meiselman, H.J. & Connes, P. (2009) Red blood cell aggregation, aggregate strength and oxygen transport potential of blood are abnormal in both homozygous sickle cell anemia and sickle-hemoglobin C disease. Haematologica, 94, 1060-1065.

Vent-Schmidt, J., Waltz, X., Romana, M., Hardy-Dessources, M.D., Lemonne, N., Billaud, M., Etienne-Julan, M. & Connes, P. (2014) Blood Thixotropy in Patients with Sickle Cell Anaemia: Role of Haematocrit and Red Blood Cell Rheological Properties. Plos One, 9, e114412.

Nebor, D., Bowers, A., Hardy-Dessources, M.D., Knight-Madden, J., Romana, M., Reid, H., Barthelemy, J.C., Cumming, V., Hue, O., Elion, J., Reid, M. & Connes, P. (2011) Frequency of pain crises in sickle cell anemia and its relationship with the sympatho-vagal balance, blood viscosity and inflammation. Haematologica, 96, 1589-1594.

- The Discussion paragraphs about DeBacker score and capillary density (lines 136- 163) could mention that the highly tortuous vessels could raise the DeBacker score. It seems that a gridline on Fig 4, two parallel vessels could touch the gridlines 4 times and one vessel with tortuous hairpin turns could also touch the gridline 4 times, with the same DeBacker score but different physiologic significance for oxygen transport and vulnerability to vasoocclusion.

Indeed, tortuosity can increase the DeBacker score but it still reflects the vessel density increase and the increase of surface exchange. We added the following paragraph in the revised manuscript: “DeBacker score encompasses arterioles capillaries, and venules. While it does not provide the same information as the "functional capillary density”, which specifically reflects the metabolic exchange potential includes arterioles, it does provide an indication of the density and perfusion of the microvascular bed (De Backer et al, 2007).”

- The Conclusion has only the briefest mention of future studies (lines 289-290).  Please consider adding to the Discussion or the Conclusion more discussion of the limitations of this novel approach and plans for future studies. For example,

  1. the ex vivo measurements provide the statistical average properties of a population of RBC, but maybe the pathophysiologic events need emphasis on the subpopulation of the most rigid cells (analogous to the large body of data that subpopulations of sickle RBC have high density vs lower density). The most abnormal sickle RBC would probably preferentially cause vaso-occlusion and so the analysis might need to include measures of the heterogeneity of the rheologic properties.

We thank the reviewer for this comment. We agree that it might be interesting to include information about RBC density heterogeneity. However, while the most dense cells have very low deformability and may mechanically cause occlusions, the less dense cells are more deformable but may adhere more than dehydrated RBCs with greater density to the endothelial cells. This greater adhesiveness to the endothelial cells may slow down blood flow to the microcirculation leading to increased RBCs transit time in the smallest capillaries that will increase the risks for HbS polymerization, RBC sickling and occlusion by the mechanically distorted RBCs. The contribution of these phenomenon is difficult to capture in-vivo and can be accessed by using microfluidic system where it is easy to follow RBCs of different densities after having isolated, washed and marked them. Indeed, at that time, we do believe that mixing in-vivo microcirculation and ex-vivo blood rheological measurements to test the clinical relevancy in the context of SCD is already very challenging.

  1. the statistical distributions of shear rate and RBC velocity are probably skewed and maybe some pathophysiologic information could be gained from examining the vessels with the lowest shear rate. The proportion of vessels with "sporadically detectable" velocities (lines 100-101) might be an important pathophysiologic measurement.

We thank the reviewer for his/her very interesting suggestion. We added several information in the revised manuscript: “Notably, in 93% ± 11% analyzed sites, no 2-4 µm vessels were flowing for the HbSS group compared to 70% ± 23% for the HbAA group (p = 0.22)” and “The prevalence of no-flow vessel in the HbSS population could reflect pathophysiologic information as this may impact perfusion/tissue oxygenation, however there was no statistical difference between the two groups, probably because of the limited sample size. Studies including more patients are clearly needed”

  1. Blood sample handling and storage conditions could influence RBC survival, so that the most abnormal RBC could be lost before ex vivo analysis.

The blood rheological measurements were always made in less than 1 hr after sampling so the risk for changes due to handling and storage conditions is very small.  Moreover, blood is reoxygeated for 15 min at room (rolling on a roller tube) before measurement. These information have been added: “Deformability and viscous measurement were performed in the hour following the sampling. The EDTA blood samples were oxygenated at room air right before ex-vivo analysis.”

  1. temporal variation in perfusion. Dr. Tom Coates group has a body of work on vasomotor regulation and analysis of the frequency of cardiovascular variations in larger vessels and heart rate. Is temporal variation also captured in the microcirculation observations by Microscan?

Microscan postprocessing software has the ability to classify vessels in categories: No flow; Intermittent; Sluggish; Continuous and Hyperdynamic . However the stabilized video must be long enough to perform the analysis which is very challenging in practice.

  1. The tongue is not a site for vaso-occlusive complications in clinical experience with sickle cell disease, so the findings . Could Microscan provide microcirculatory imaging in any sites that do have vasoocclusion?

We agree that the tongue is not a site for vaso-occlusive complications. The focal point of the device is specifically calibrated for sublingual microcirculation and the companies of the microscan did not develop other systems that could be used in other sites. To be used on another site, an explorative and a validation study must be performed.

line 44 "Although gene therapy is" should be "Although gene therapy and hematopoietic stem cell transplant are"

The corrections has been done.

line 94 "repartition of the vessel diameters" might be more clearly termed "distribution of the vessel diameters".

The corrections has been done.

lines 197-198 - please specify the storage time, oxygen conditions, and temperature before ex vivo analysis, and whether the blood sample remained in EDTA during that time.  These parameters can influence sickle RBC

The information has been added: “Deformability and viscous measurement were performed in the hour following the sampling. The EDTA blood samples were oxygenated at room air right before ex-vivo analysis.”

line 218-219  - please specify the oxygen conditions for the cone/plate viscometer. Presumably it is room air.

The reviewer is right. We added this information.

line 241-242 "video microscopy recordings of three to six sublingual sites were analyzed." What determined the number of sites - quality of image? number of datapoints? the ability of the volunteer to hold the tongue position?  what was the duration of recording - could it capture temporal variation of vasoregulation?

Yes, the quality of the videos are the criteria. It is difficult for patients to not move the tongue and in addition to the operator maintaining the head with a constant pressure and with the vessels on focus require dexterity and practice. The videos are few seconds long for each site. During post-processing each video was explored to find a sequence of a couple of seconds on focus that could be stabilized.

We added the following paragraph in the revised manuscript:

“The sites analyzed were determined by the video quality obtained as patients found it difficult to hold their tongue still. Furthermore, to avoid data inaccuracies, the operator maintained constant pressure on the Microscan handheld device, which requires dexterity and practice. The video for each site was approximately 10 seconds long, from which the most stable sequence of frames was extracted for analyses.”